# *Mycobacterium leprae*’s Infective Capacity Is Associated with Activation of Genes Involved in PGL-I Biosynthesis in a Schwann Cells Infection Model

**DOI:** 10.3390/ijms24108727

**Published:** 2023-05-13

**Authors:** Bibiana Chavarro-Portillo, Carlos Y. Soto, Martha Inírida Guerrero

**Affiliations:** 1Hospital Universitario Centro Dermatológico Federico Lleras Acosta, Avenida 1ra # 13A-61, Bogotá 111511, Colombia; bchavarrop@unal.edu.co; 2Chemistry Department, Faculty of Sciences, Universidad Nacional de Colombia, Ciudad Universitaria, Carrera 30 N° 45-03, Bogotá 111321, Colombia; cysotoo@unal.edu.co

**Keywords:** Schwann cells (SCs), infection, *Mycobacterium leprae* (*M. leprae*), leprosy recurrence, phenolic glycolipid-I (PGL-I), intracellular pathogen

## Abstract

Peripheral nerves and Schwann cells (SCs) are privileged and protected sites for initial colonization, survival, and spread of leprosy bacillus. *Mycobacterium leprae* strains that survive multidrug therapy show a metabolic inactivation that subsequently induces the recurrence of typical clinical manifestations of leprosy. Furthermore, the role of the cell wall phenolic glycolipid I (PGL-I) in the *M. leprae* internalization in SCs and the pathogenicity of *M. leprae* have been extensively known. This study assessed the infectivity in SCs of recurrent and non-recurrent *M. leprae* and their possible correlation with the genes involved in the PGL-I biosynthesis. The initial infectivity of non-recurrent strains in SCs was greater (27%) than a recurrent strain (6.5%). In addition, as the trials progressed, the infectivity of the recurrent and non-recurrent strains increased 2.5- and 2.0-fold, respectively; however, the maximum infectivity was displayed by non-recurrent strains at 12 days post-infection. On the other hand, qRT-PCR experiments showed that the transcription of key genes involved in PGL-I biosynthesis in non-recurrent strains was higher and faster (Day 3) than observed in the recurrent strain (Day 7). Thus, the results indicate that the capacity of PGL-I production is diminished in the recurrent strain, possibly affecting the infective capacity of these strains previously subjected to multidrug therapy. The present work opens the need to address more extensive and in-depth studies of the analysis of markers in the clinical isolates that indicate a possible future recurrence.

## 1. Introduction

Leprosy is a non-traumatic peripheral neuropathy whose causal agents are *Mycobacterium leprae* and *Mycobacterium lepromatosis* [1,2]. This generalized peripheral neuropathy is a chronic infection that produces inflammatory lesions on the skin and in peripheral nerves [3], which comprises a complex spectrum of clinical and immunological manifestations that range from localized forms imperceptible to the patient—indeterminate leprosy—to disseminated forms involving peripheral nerves, deformity, and disability. The clinical manifestations depend on the specific immunological reaction of the patient against the bacillus and are classified into two polar forms: lepromatous leprosy (LL), a predominantly humoral response, and tuberculoid leprosy (LT), a predominantly cellular response [4].

*M. leprae* is an obligate intracellular pathogen with an exquisite tropism for Schwann cells (SCs), glial cells of the peripheral nervous system (PNS) derived from neural crest precursors. *M. leprae* shows remarkable plasticity and is sensitive to changes that may affect its functions; in addition to lacking antimicrobial activity, the SCs can tolerate a high bacterial load [5,6]. The protection given by the SCs to *M. leprae* during its multiplication has allowed bacterial persistence for long periods, since the resident bacillus in the SCs serves as a primary source of infection that, in addition to causing nerve damage, facilitates dissemination to other organs and tissues [7,8,9].

The tissue tropism of the bacterial pathogen is determined by both the host and the bacterial components. In pathogenic mycobacteria, it has been proposed that their bacterial wall contains most of the elements associated with the disease pathogenesis [10,11]. *M. leprae,* like other pathogenic mycobacteria, has specific pathogenic components in a bacterial wall, such as phenolic glycolipid 1 (PGL-I), a critical molecule for *M. leprae* internalization in SCs, and because it is found in the outer layer of bacillus, it is ideal for interaction with host cell components [12,13]. PGL-I contains an antigenically distinct trisaccharide consisting of 3,6-di-o-methyl-β-D-glucopyranosyl-(1→4)-2,3-di-o-methyl-α-L-rhamnopyranosyl-(1,2)-3-o-methyl-α-L-rhamnopyranose, antigenically unique and not found in any other mycobacterial species or general bacterial species [12,14,15]. PGL-I is involved in the unique affinity of *M. leprae* for peripheral nerves. The PGL-I biosynthetic pathway involves more than 20 enzymatic steps; the lipid core formation is typical in all PGL-containing mycobacteria, but the saccharide portion of PGL is specific to each mycobacteria [16]. The trisaccharide position of PGL-I has been proposed to promote SC invasion and thus may be responsible for the unique ability of *M. leprae* to invade peripheral nerves [5,12,17].

Although the application of multidrug therapy (MDT) has reduced cases of leprosy worldwide, every year there are reports of disease recurrence events in patients who completed MDT, which may be due to the persistence of the bacillus in the tissues [18,19]. The growth of surviving bacilli, or those in the latent phase within the SCs, can occur even some years after the termination of MDT [18,20]. This situation can occur as a result of the activation of the metabolic state of the mycobacterium that was inactive or due to the persistence of live bacilli in the peripheral nerves, lymph nodes, or skin [20,21,22]. Factors influencing recurrence include delayed initial diagnosis, high initial bacillary loads, and leprosy in multibacillary patients [18]. This clinical manifestation differs from processes such as drug resistance and reinfection, the latter in patients who have been cured and years later start the disease again with a new strain of *M. leprae* [23,24].

*M. leprae* is not cultivable in artificial media, so its multiplication has been restricted to animal models, especially in the footpads of mice; based on this model, the bacillus has been obtained to carry out SC infection assays [25]. This methodological approach has allowed the establishment of factors involved in the interaction between the bacillus and SCs, intracellular survival, and dissemination of the bacillus, among others [13,26,27]. Obtaining *M. leprae* from the mouse footpad assay is not commonly used due to the specific infrastructure requirements for handling mice and cell cultures. Ideally, the primary source of *M. leprae* for experimentation in cell models such as SCs would be bacilli from patients with active leprosy. However, this approach also presents difficulties in its development, which is why it is rarely used. New developments include recovering bacilli directly from biopsy samples or taking interstitial fluid for disease diagnosis and purification for cells infection. It is for the above that the present work aims to determine the infection capacity and its relationship with the transcription levels of genes involved in the biosynthesis of PGL-I from *M. leprae* clinical isolates that cause recurrent events of leprosy in a Schwann cell infection model.

## 2. Results

### 2.1. Mycobacterium leprae Purification from Biopsy Samples of Patients with Recurrent or Non-Recurrent Leprosy Events

Five multibacillary (MB) patients diagnosed with LL (4 with non-recurrent and 1 with recurrent leprosy) who had bacteriological indices (BI) ranging from 1.5+ to 5+ were included in this study. The samples of new cases presented clinical signs of the disease between 7 and 48 months before diagnosis. The recurrent case’s sample came from a patient who, 40 months before, had completed MDT. The appearance of new clinical signs and the recurrence diagnosis included positive bacillary viability and negative test to drug resistance determined by mutations in the drug resistance-determining region (DRDR) for the *folP1*, *rpo β*, and *gyrA* genes from *M. leprae* [24].

Different bacillary concentrations of suspension were obtained from the fresh biopsies, variations that were expected to be found due to the intrinsic characteristics of each patient and are related to their genetic repertoire, immunological response, and the disease stage (recurrence or non-recurrence) (Table 1).

### 2.2. Morphological Description of Schwann Cells Infected with M. leprae from Recurrent and Non-Recurrent Events

For a successful entry of *M. leprae*, a pathogen with an exquisite tropism for SCs, these cells must have a homogeneous structure that allows the entry of *M. leprae* and its multiplication and viability in addition to facilitating the infection of more cells in the long term. Primary human Schwann cell lines (PHSC) (ScienCell) were infected with *M. leprae* clinical isolates from patients with recurrent or non-recurrent leprosy and remained in culture for 3, 7, 10, and 12 days after infection (Figure 1).

Through Zielh Neelsen staining, intracellular bacilli were confirmed at all time points evaluated, and changes in the morphology of infected versus uninfected PHSCs were observed at these time points (Figure 1). At 3 days post-infection, intra-cellular bacilli were observed that followed the direction and surface given by the cytoplasm of the cells. This evidences cell culture’s purity, congruence, and richness in the well. At 7 days, a cellular alteration was observed in the PHSC, characterized by cytoplasm contraction and increased vacuolization/lipid droplets in the infected cells. Finally, at 12 days of infection, notable cell loss was observed. Most cells show bacilli at the cytoplasm level and with a characteristic contraction of the same, which is also evidenced by the decrease in the monolayer over time.

### 2.3. Infection Kinetics of Schwann Cells with Different Clinical Isolates of M. leprae

As the infection model allowed *M. leprae* entry into PHSC, the infective capacity of each clinical isolate was evaluated by determining the proportion of infected cells over time. Significant changes were observed in the proportion of infected cells between each case analyzed (non-recurrent vs. recurrent) (Figure 2, Table 2). The infectivity of *M. leprae* clinical isolates from patients with non-recurrent events is always more significant; a median increase of 12.83% to 18.5% was observed between days 0 and 3 in all cases and remained constant until day 12, reaching a maximum of 52.5% infection.

The recurrent clinical isolate showed a lower capacity of infection compared with the non-recurrent clinical isolates. At Day 0 to Days 3, 7 and 12 post-infection, the percentage of cell infection was 6.5%, 8.5%, and 22%, respectively. In the case of dead bacilli (*M. leprae* Thai-53), cell infection reached 22%, which remained constant over time (Appendix A).

### 2.4. Replication Kinetics of M. leprae in Schwann Cells

Figure 3 shows the average number of bacilli by the cell in each experiment at each time analyzed. In the study, clinical isolates showed differences in multiplication capacity over time. On Days 3 and 7, the clinical isolates of the experiments that used live bacilli showed a progressive increase in the number of bacilli per infected cell, indicating that from this day on, *M. leprae* multiplied in the model used (Table 2).

The number of bacilli per infected cell of non-recurrent clinical isolates increased at Day 3 (1.37 to 3.38 bacilli per cell) and Day 12 (1.744 to 3.24 bacilli per cell), (*p* ≤ 0.0001) (Appendix A). In the case of the recurrent clinical isolate, the increase in the number of bacilli per infected cell occurred on Day 7 (1.71 bacilli) compared with bacilli found on Day 3 (1.18 bacilli per cell) (*p* ≤ 0.0001). The multiplication control corresponds to dead bacilli of the Thai-53 strain, which do not show changes in the number of bacilli per cell (*p* ≥ 0.9999), indicating no multiplication of these bacilli. The results indicated a variation in infective bacilli in each analyzed time.

### 2.5. Relationship between Multiplication Capacity and Viability of M. leprae

The increase in the number of cells infected by *M. leprae* over time depends on the viability of the bacillus long enough to infect new cells; Table 2 shows the average number of infected cells per well and the average number of infecting bacilli per well at each of the times evaluated. Since the number of bacilli increased at 3 days post-infection in non-recurrent events and at 7 days post-infection in recurrent events and was maintained over time, the evaluation of bacterial viability correlated with the normalized expression of the *sod* gene.

Regarding non-recurrent bacilli, a 10-fold increase average in the quantification of the transcription of the *sod* gene on Day 3 post-infection compared with Day 0 (Figure 4) was observed. In the case of recurrent clinical isolates, the *sod* gene transcription increased 5-fold between Days 3 and 7 post-infection. Thus, the percentage of infection and the multiplication capacity of *M. leprae* in the infection model agreed with the levels of *sod* gene transcription, confirming the viability of the bacillus from patients.

### 2.6. Level Transcription of Genes Involved in PGL-I Synthesis during Schwann Cell Infection with M. leprae Clinical Isolates

The genes for transcriptional analyses were selected for their pivotal role in PGL-I biosynthesis [16]. Concerning the recurrent clinical isolate, the *ppsC* gene (involved in the synthesis of the phenolphthiocerol/phthiocerol portion (Figure 5a,b) showed a relative transcription of 0.76 at 2 h post-infection in the recurrent clinical isolate, while the level transcription of *ppsC* increased from 1.7-fold on Day 3 to 8.27-fold on Day 10 and showed a maximum peak at 7 days post-infection (Figure 5c).

For the non-recurrent clinical isolates, at 2 h post-infection the *ppsC gene* showed an average relative transcription of 9.08, which increased 29-fold on Day 3 post-infection, with a subsequent decrease on Days 7 and 12 post-infection (*p* > 0.0001). When the level transcription was compared between *M. leprae* clinical isolates depending on their origin, non-recurrent *M. leprae* increased transcription 30-fold on Day 3. In contrast, recurrent *M. leprae* increased transcription 6-fold on Day 7 (Figure 5c). As observed for the *ppsC* gene, the transcriptional level of the ML2348 (coding for glucosyltransferase), ML2346c, and ML2347 (coding for methyltransferases) genes was upregulated in non-recurrent events, which could be partially due to those genes being part of the same operon at the same locus (Figure 5a,b,d–f).

The analysis of the transcriptional behavior of the ML0126, ML0127, and ML0128 genes, located in a different locus in the *M. leprae* genome (Figure 5g), which are involved in the process of methylation and translocation of the rhamnosyl residues of PGL-I (Figure 5b), were intriguingly compared with genes at locus 1 (ML2346c, ML2347, ML2348). Gene transcription of ML0126 and ML0127 was activated in recurrent *M. leprae*, specifically ML0126 between Days 0 and 7 post-infection (Figure 5h,i). For the non-recurrent *M. leprae*, the ML0126 and ML0127 genes were activated on all the days analyzed, reaching their maximum peak at 3 days post-infection, which decreased over time. The ML0128 gene began with a negative transcription at 2 h post-infection, rising at 3 days and decreasing negatively at 7 days post-infection (Figure 5j). Since the function of glycosyltransferases during the synthesis of PGL-I may be assumed by the ML2348 gene (locus 1), ML0128 does not have a significant action, especially in clinical isolates from non-recurrent events.

## 3. Discussion

The most severe clinical sequela of leprosy is the development of neuropathies that result from an infection of SCs by *M. leprae*. In multibacillary leprosy, there is a high number of intracellular bacilli, producing axonal degeneration that generates not only disability but also bone loss and muscle atrophy secondary to loss of sensitivity, which is also related to complications that affect the adjacent tissues of the skin [3,28,29]. These conditions have determined that leprosy remains one of humanity’s most feared and stigmatized diseases [30].

A major obstacle to laboratory research on *M. leprae* is associated with the difficulty of multiplying the bacillus in cell culture and its slow doubling time (∼14 days) [25]. The capacity for interaction, invasion, and subsequent multiplication of *M. leprae* in SCs is a process that has been investigated on numerous occasions using clinical isolates of *M. leprae* from animal models. One of them is the nine-banded armadillo (*Dasypus novemcinctus*) and mouse footpads (MFP) [31,32,33], models that allow the multiplication of the bacillus due to an inadequate immune response against infection by *M. leprae*. These studies included significant numbers of bacilli to investigate cellular, biochemical, and molecular aspects [13].

After the application of MDT in recent years, an increase in the number of relapses has been observed. For 2016, 52 countries reported 2844 relapses out of 217,971 new cases; in 2017, 3192 cases out of 211,182; and in 2018 and 2019, 3361 and 3893 cases out of 208,641 and 202,185 new cases, respectively [34,35]. When reviewing the worldwide incidence of relapse cases in leprosy, it is clear that it is low (1.16 per 1000 py (95% CI = 0.5915–2.076)) [18,36], that it is necessary to follow 1000 leprosy cases to obtain less than two cases of relapse of the disease, and that the period of follow-up could be long and variable because of censoring and loss to follow-up. This makes it challenging to include more cases of leprosy relapses in the present study, [36,37] especially considering that Colombia has 400 new cases of leprosy per year [34]. This work is a descriptive observational study that was not intended to extrapolate national or international data. Furthermore, given the characteristics of the disease, the inclusion of at least one case of leprosy relapse provides us with essential data that deserves to be analyzed and socialized.

The present study compared the infective capacity of *M. leprae* clinical isolates extracted and purified from fresh tissue of patients with initial leprosy untreated to bacilli from patients with recurrent events of leprosy. Different amounts of the bacillus was isolated from the biopsy samples, depending on the clinical, genetic, and immune characteristics of patients, thus producing different BI (r = 0.919), which was estimated when the biopsies were obtained from donor patients (Table 2). However, these amounts (BI), used as infective suspensions in our experiments, are not related to the percentage of infected cells in the T0 of experiments, which may be related to the inherent characteristics of each clinical isolate (Table 1).

The quantities of bacilli obtained allowed us to adapt a multiplicity of infection (MOI) of 10:1, unlike what occurred with previous studies [38]. The conditions used in this study allowed the cultures to be kept infected for up to 12 days, with no monolayer detaching due to dehydration or the action of catabolic products from *M. leprae*. When the percentage of infection of the SCs with clinical isolates of *M. leprae* from patients with recurrent and non-recurrent events was evaluated, a differential behavior was found, showing that *M. leprae* from recurrent patients increases its infective capacity at 7 days post-infection compared to *M. leprae* from non-recurrent patients, which increases steadily and sustains earlier, from Day 3 post-infection. The initial interaction time between the bacillus and the cells was only 2 h. After this time, the cells were washed four times with PBS 1X to remove the bacteria in the supernatant that was not internalized by the cell, which guaranteed that subsequent bacillary load quantifications over time correspond to actual infections and the increase in the number of bacilli (Table 2).

An essential feature of mycobacteria is their ability to adapt to stress conditions, such as the presence of antibiotics, an active immune response, and permanence in the host cell. The lag phase is characterized by growth arrest or retardation, anaerobic respiration, and tolerance to antibiotics [39]. Under physiological conditions, bacteria in a latent state are contained mainly in granulomas, cells that protect them from the action of the immune system [40]. In our studies, the results of the infective capacity among the clinical isolates evaluated show that the clinical isolates from recurrent events require a longer adaptation time to the infection environment, so we can speculate that they were in a state similar to latency, reflected by decreased metabolism [41,42].

By Day 12, it was observed that cellular infection varied between 13% and 54% (Figure 2 and Table 2); this result shows that the SCs can reach between 80% and 100% infection with the leprosy bacillus, according to the results presented by Diaz Acosta et al. [13]. When we used samples with dead *M. leprae*, a 26% infection of SCs was reached, a result that also was observed by Diaz et al., in which it was established that the percentage of SCs with dead *M. leprae* bacilli decreased to less than 50% compared to that achieved by viable bacilli. This may be because upon the death of the bacillus, the structures of the cell wall may be altered so that the interaction with the host cell is affected [13,43].

The present work also evaluated the dynamics of the bacillary load over time, that is, the number of bacilli per infected cell (Figure 3 and Table 2). Our results showed that the number of infective bacilli does not depend on the number of bacilli in suspension obtained from the original sample used for infection (r = 0.295) and that the viability of the bacillus is not required for the initial entry of *M. leprae* to the SCs but is essential for the increase of the infection or bacillary load over time, similar to the results obtained in previous studies [13,43,44]. Assuming that our infectious populations are asynchronous, as they come directly from the patient, we cannot assume that the population increase has doubled in terms of generation time (g), which is a period of 14 days. However, Tukey’s analysis shows that as in Figure 3 and Table 2, the increase in population at these points in the experiment was significant, with an increase in the total population per well being achieved from 28,180 on Day 0 to 34,200 on Day 7 from the clinical isolates of recurrent events, while for clinical isolates of non-recurrent events, these data were 45,200 to 51,600, respectively. These results were confirmed with the analysis of the viability that *M. leprae* showed during its multiplication in the SCs, which demonstrated that the experimental model used allowed us to not only maintain viability but also stimulate the multiplication of the *M. leprae* clinical isolates coming directly from the patient, this being the first study to perform in vitro analysis of *M. leprae* from clinical samples.

For our study, bacillus viability estimated by *sod* gene transcription was related to the number of bacilli per cell and increased as the bacillus multiplied in the infection model, in which at least 10,000 bacteria were used at Time 0 (2 h) [45]. The count of the bacilli and the determination of the viability showed that the bacteria are capable of multiplying, even in the case of *M. leprae*, causing recurrent events, possibly reduced due to the previous application of antibiotic treatment in the patient. Previous studies have shown that once antibiotic treatment is withdrawn, intracellular mycobacteria resume their growth immediately. After a few days, their replication rate was identical to that of bacteria that had not been previously treated [46]. It is essential to highlight that in this case, *M. leprae* is the bacteria’s longest generation time (∼14 days); therefore, it could require a longer adaptation time to restart its post-treatment growth.

The ability of *M. leprae* to bind through a specific interaction of PGL-I with the alpha-2 subunit of human laminin (LAMA2) promotes the attachment of the bacillus to the basal lamina of the SCs. This allows the internalization of the pathogen [12,17,47]. In addition, *M. leprae* has developed PGL-I production to evade innate immunity and establish long-term residence in the host [16]. In this study, the possible participation of genes involved in the synthesis of the trisaccharide component of PGL-I in the infective capacity of *M. leprae* from clinical samples was evaluated. Since it is known that intracellular pathogens such as *M. leprae* require the lipids of the host cell to ensure the successful colonization of the microorganism and the progression of the infection, specially PGL-I, the measurement of transcription of genes involved in the biosynthesis of this glycolipid is a reliable way to estimate the multiplication and infective capacity of *M. leprae* at the intracellular level [48]. This should be valid in clinical isolates of *M. leprae* without previous exposure to antibiotics (non-recurrent) and for microorganisms in a state of latency or recurrence.

The PGL-I biosynthetic pathway involves more than 20 enzymatic steps, including the synthesis of the lipid core common in mycobacteria. Specifically, the genes involved in the synthesis of the trisaccharide portion of PGL-I are ML0128, which codifies a rhamnosyl transferase of the second rhamnosyl residue in position 2 of the first glycosyl residue; ML0127, which codifies a methyltransferase involved in the methylation of position 2 of the second rhamnosyl residue; and ML0126, which codifies the enzyme responsible for methylation at position 3 of the first and second sugar residues. Regarding the transfer and modification of the terminal glycosyl residue, ML2348 codifies the glycosyltransferase and ML2346c, and ML2347 codifies the methyltransferases required for the modification of positions 3 and 6 [16]. The mentioned genes grouped in two regions of the *M. leprae* genome have orthologs involved in developing PGL and thiocerol dimicoserosate of *M. tuberculosis* [16,49]. On the other hand, the *ppsC* gene synthesizes the phenol-thiocerol portion of PGL-I. Suppose that transcription levels of *ppsC* (>30-fold) are activated in clinical isolates of non-recurrent events. In that case, it is strongly suggested that these clinical isolates contain higher levels of PGL-I, which have been associated with the pathogenicity of *M. leprae*; since this glycolipid is unique to *M. leprae* and is highly antigenic, it is therefore not found in other mycobacteria [12,50]. This is consistent with the increased infection kinetics of clinical isolates from the non-recurrent event observed in SCs.

We also compare the transcriptional behavior of the genes involved in the synthesis of the saccharide core of PGL-I, which are distributed into two different loci. In Locus 1, we found that ML2348 (glycosyltransferase), ML2346c, and ML2347 (methyltransferases) required structural modification of the sugars. We found a differential behavior between transcription levels of the clinical isolates of *M. leprae* from recurrent and non-recurrent events that were higher on Day 3 post-infection (*p* < 0.0001) for non-recurrent events and at Day 7 post-infection for the recurrent event (*p* = 0.8867) and, accordingly, with transcription levels of ML2348, ML2346c, and ML2347, indicating that the synthesis of PGL-I could increase in those times. Therefore, the interaction of *M. leprae* with the host improves, producing an increased number of infected cells. At Locus 2, we analyzed gene transcription ML0128 (rhamnosyl transferase), ML0126, and ML0127 (methyltransferase). In this case, we observed a different behavior from that observed in Locus 1 because the ML0128 gene showed a transcript less than 1 in recurrent *M. leprae*. In the case of the non-recurrent clinical isolates, the ML0128 gene only presented a transcript greater than 1 at Day 3 post-infection. The function of this glycosyltransferase could be supported by another involved glycosyltransferase (ML2348), which is found in Operon 1. The complete transcription analysis suggests that the rate of synthesis of the PGL-I, essential for the entry of *M. leprae* into SCs, is different in the two types of clinical isolates, explaining why the recurrent event of the disease occurs on average 7 years after the first cure post-MDT and why the causative bacillus could be found in a state of latency [18].

These differences are related to the ability of *M. leprae* to limit its rate of replication in extreme conditions; a recovery of the replicative state occurs when favorable conditions return [51]. In our case, the strain of *M. leprae* from the recurrent case required a longer adaptation time to increase its replication rate and infectivity when compared to a strain that had not been previously exposed to antibiotic treatment. Another aspect that contributes to the growth kinetics and transcription genes involved in the biosynthesis of *M. leprae* PGL-I is the genetic and immune background of the host cell. Previous studies found that IFN-β secretion by macrophages correlated with the multiplicity of infection by *M. leprae* in vitro [52]. Although the induction of IFN-I is an essential factor for controlling viral infections, it has been shown to suppress the antibacterial response, particularly with infections by intracellular bacteria [52,53].

## 4. Materials and Methods

### 4.1. Ethics Statement

This study was approved by the ethical committee of Hospital Universitario Centro Dermatológico Federico Lleras Acosta (HUCDFLLA) (Empresa Social del Estado/Ministerio de Salud y Protección Social, Colombia). Informed consent was obtained from all adult volunteer patients before inclusion in the study (study identifier code: 4000-16.2H), considering the ethical standards in medical research according to the Declaration of Helsinki, where it is stipulated that it is a risk-free investigation.

### 4.2. Collection and Processing of Clinical Material

Five target patients were classified clinically, bacteriologically, and histopathologically according to the Ridley–Jopling scale [4]. Inclusion criteria for recurrent cases previously treated with MDT (and followed up at HUCDFLLA), included negative drug-resistance evaluation and positive bacillus viability; in addition, the diagnosis of recurrence was confirmed with a positive biopsy. In the cases of non-recurrent events (new cases), these should not have received prior treatment. From these patients, five elliptical incisional biopsies were obtained with the following dimensions: 1 cm long × 5 mm wide × 7 mm deep and performed by the dermatologist responsible for Hansen’s clinical at HUCDFLLA, Bogotá, Colombia.

### 4.3. Purification and Quantification of Mycobacterium leprae

The biopsies of the patients were transferred to a sterile petri dish, and with the help of a scalpel, the tissue was sectioned. The sectioned material was transferred to a sterile glass tissue homogenizer (Ref 7726-L, PYREX^®^, Corning, Tewksbury MA, USA) and macerated, keeping the homogenizer on ice. Once the tissue had been macerated, 1 mL of cold Hank’s balanced salt solution (SSBH) (Ref H4641-100ML, Sigma, Irvine, UK) was added and homogenized into a suspension and filtered through a Falcon^®^ 40 µm Cell Strainer (Ref 352340 BD). The cell debris that remained in the filter was taken and placed again in the homogenizer. An amount of 1 mL of cold SSBH was added, and the tissue was homogenized again, and the suspension was then filtered again and once more until reaching a volume of 2.5 mL. An amount of 500 µL of 0.5% trypsin ([0.05%] final) was added to the suspension, and it was incubated in a water bath at 37 °C for 1 h. Subsequently, 10 mL of SSBH was added to the suspension and centrifuged at 3500× *g* for 30 min at 4 °C. Once the supernatant was discarded and 1 mL of SSBH was added to the precipitate, the suspension was homogenized [26,31].

Bacillary quantification was performed according to the criteria established by Shepard and McRae (Shepard and McRae, 1968). Briefly, the number of acid-fast bacilli (AFB) was counted by direct examination of 20 fields at 100× (immersion oil) of each of the three circles of 1 cm in diameter, each containing 10 μL of suspension bacillary, checking along the horizontal axis of the stained smear and using a microscope with a calibrated objective. The mean number of bacilli in each of the three smears was determined and multiplied by the appropriate calibration factor to obtain a mean and standard deviation for the AFB count. Care was taken to quantify only intact and fully stained bacilli. Bacillary viability was confirmed using the BacLight LIVE/DEAD viability kit (L7007, ThermoFisher Scientific, Eugene, OR, USA), according to the manufacturer’s instructions.

### 4.4. Human Schwann Cell Culture

Primary human Schwann cell lines (PHSC) were obtained from the human neural crest derivatives that ensheathe and myelinate axons of peripheral nerves and were purchased from ScienCell, Carlsbad, CA, USA (Cat No. 1700). The PHSC was maintained in Schwann cell medium (SCM, Cat. no. 1701, ScienCell), supplemented with Schwann cell growth supplement (SCGS, Cat. no. 1752, ScienCell) in cell culture flasks previously coated with Poly-L-lysine 2 µg/cm^2^ (Cat. No. 0413, ScienCell) to promote cell adhesion, according to the suppliers’ recommendations. The cells were cultivated at 37 °C with 5% CO_2_, and the culture medium was renewed every 3–4 days, with two previous washes with PBS 1X.

The cells were washed with PBS 1X twice and subjected to a monolayer separation process with a Trypsin/EDTA solution at 0.0625% (ScienCell) for 1 min. Once the cell separation was verified in the optical microscope, the cells were collected in a 50 mL tube containing Bovine Fetal Serum (FBS); 5 mL of neutralization solution (ScienCell) was added, and then the mixture was centrifuged at 3000× *g* for 5 min. Once centrifuged, the count and evaluation of cell viability (in a Neubauer chamber) were carried out. An amount of 10 μL of the cell suspension was taken and mixed with 90 μL of trypan blue (1:10 dilution); 10 µL of the dilution was placed in a Neubauer chamber, and the cells were observed under a light microscope and counted in the four grids for white blood cells. The number of cells/mL was calculated as the average in the four quadrants × dilution factor × 10,000. The cell suspension was then prepared in supplemented SCM medium, according to the number of cells required for the infection assay (20,000 cells per well). Cells were cultured in 24-well plates on slides at the bottom of the well and were previously treated with Poly-L-lysine.

### 4.5. Schwann Cell Infection with Clinical Isolates of M. leprae

Once the SCs were in a monolayer on a slide, in 24-well plates (Falcon, Franklin Lakes, NJ, USA), at a density of 20,000 cells per well, the previously quantified bacillary suspensions of each case were added in a ratio of 10 bacilli per cell (MOI = 10:1). The samples were allowed to interact at a temperature of 33 °C with 5% CO_2_. The initial interaction time between the bacillus and the cells was 2 h only. After this time, the cells were washed four times with PBS 1X to remove the bacteria in the supernatant that was not internalized by the cell. Infected cells were washed every 72 h before adding an SCM-supplemented medium to avoid dehydration and alkalinization and were kept at a temperature of 33 °C with 5% CO_2_.

Once the incubation time of SCs infected with *M. leprae* was completed—2 h (T0 = 2 h), 72 h (T1 = 3 days), 7 days (T2 = 7 days), 10 days (T3 = 10 days) and 12 days (T4 = 12 days) in the 24-well plates—the medium was discarded, and the cells were washed twice with 1X PBS. Cells were fixed by adding 4% paraformaldehyde and incubated at 37 °C for 10 min. Once the paraformaldehyde had been discarded, the cells were washed twice with 1X PBS (between each wash, they were incubated at room temperature for 5 min). Then, the slides were removed from the well, and the slides were stained using the Zielh Neelsen stain.

To establish the kinetics of infection, the previously fixed slides were covered with filtered carbon fuchsin, and the dye was left in contact with the monolayer for 30 min. After the fuchsin was discarded, the slides were washed with water; then, 3% acid alcohol was added for 5 min, and the slides were washed again with water. Subsequently, the slides were covered with Loeffler’s methylene blue (contrast dye) for 20 min, washed with water, and dried at room temperature. Mounting was then carried out with a cytoresin around the slide, leaving the study sample in contact with a slide to preserve the previously stained sample.

The infected cells were checked and quantified by the number of bacilli per cell by counting the proportion of 200 infected SCs and the number of bacilli in each cytoplasm using the light microscope. As a control, uninfected cells and other cells infected with dead bacilli from the mouse plantar pad were used, kindly supplied by Patricia Sammarco Rosa, Ph.D., of Laboratory Animal House, Instituto Lauro de Souza Lima (ILSL), Brazil.

### 4.6. Viability Assays of Mycobacterium leprae Infecting SC Culture

The infected cells were collected at 0, 3, 7, 10, and 12 days post-infection and were stored at −80 °C until RNA extraction by mechanical lysis. On the day of the experiment, the cell suspension was centrifuged at 15,000× *g* for 30 min at 4 °C. After mechanical lysis, we used the Qiagen RNeasy kit per the manufacturer’s instructions. The isolated RNA was stored at −80 °C until use. The quality (260/280 nm) RNA concentration was measured in a NanoDrop 1000TM equipment (Thermo Scientific, Wilmington, DE, USA).

The normalized *sod* gene was used to stablish *M. leprae* viability. Briefly, the extracted RNA was treated with DNAse, and cDNA was obtained by reverse transcription using M-MLV RT 200 U/µL and random hexamers. Transcription of the *sod* gene was assessed using primers *Sod-F*–CACCGTTCGGAGAGAGGTTC- and *sod-R*-TCAACGAGATCCACCACACC (Table 3). To verify the viability of *M. leprae* in this assay, a standard curve was used to extrapolate viability with the number of bacilli per infected cell. Two biological replicates for infection assays were performed.

Serial dilutions of clinical isolates of *M. leprae*’s that infected Schwann cells were pooled and lysed, and cDNA was prepared to qPCR run for the *sod* gene. The CT values of the *sod* gene were obtained from the clinical isolates of recurrence, non-recurrence, and death bacillus, and their difference were calculated. DCT = CT (recurrence or non-recurrence) − CT (dead). These ΔCT values were plotted against the number of bacilli per well to create a semi-log regression line.

### 4.7. qRT-PCR of Genes Involved in PGL-I Biosynthesis

The transcription of genes involved in PGL-I biosynthesis was quantified by real-time PCR (qPCR). For this purpose, specific oligonucleotides were designed for the selected genes (Table 1) using the primer-Blast [54] and Primer3Plus tools (Table 3) [55]. In the amplification reactions, SYBR Green was used as a fluorescence generator in a T100TM Thermal Cycler (BioRad).

All samples of total RNA were reverse-transcribed using reverse transcriptase (M-MLV RT 200 U/µL) (Thermo Scientific) per manufacturer recommendations. A mix of Oligo(dT) and random primers was used as the priming agent, and samples were stored at −20 °C until use.

For the relative quantification of the cDNA by qPCR, the commercial kit *sso*Advanced Universal SYBR Green Supermix kit from BioRad^®^ was used, following the manufacturer’s recommendations, using 10 ng of cDNA for quantification reactions. Reactions were supplemented with oligonucleotides at a final concentration of 0.25 μM (*sod*, ML0126, ML0127, ML2348), 0.375 μM (ML2346, ML2347), and 0.5 μM (ML0128 and *ppsC*). The thermal cycles included an initial denaturation at 95 °C for 5 min, 40 cycles of denaturation at 95 °C for 15 s, and an extension at 62 °C for 30 s. All experiments were performed in triplicate on two experimental replicates. The *sod* gene (superoxide dismutase) was used as an endogenous control. The negative expression values of the genes were obtained by converting the cycle threshold (Ct) values according to the following formula: Expression value = 2^(−ΔΔCt)^ DCT = CT (recurrence or non-recurrence) − CT (dead). These ΔCT values were plotted against the number of bacilli per well to create a semi-log regression line [56].

### 4.8. Statistical Analysis

The results were analyzed in terms of mean ± standard error of the mean (SEM) of independent experiments in triplicate. Data were analyzed by applying a one-way or two-way multi-measurement analysis of variance (ANOVA) with Tukey’s test. Numerical data were analyzed using the non-parametric Mann-Whitney test. Using GraphPrims version 9 for macOS Big Sur, GraphPad Software, San Diego, CA, USA, www.graphpad.com.

## 5. Conclusions

The results of the present study of human SC infection demonstrated that the ability of *M. leprae* to multiply under the optimal conditions used depends on the microbiological characteristics of each clinical isolate as well as possibly on the previous exposure of the bacillus to drugs of MDT; this can induce a modification in the metabolism of the bio-synthetic pathways of crucial molecules such as PGL-I, specifically, for clinical isolates from recurrent events. In this study, for the first time two different behaviors at the cellular and transcriptional levels were investigated and found between the clinical isolates causing recurrent and non-recurrent events, so it is necessary to delve into the transcriptional studies of *M. leprae* clinical isolates coming directly from the patient to guide their management against the possible prognosis. Therefore, the present work opens the need to address more extensive and in-depth studies of the analysis of markers in the clinical isolates that indicate a possible future recurrence. This means that patient monitoring is necessary until treatment completion and a more prolonged post-MDT clinical follow-up to guarantee that no new symptoms have appeared.

## Figures and Tables

**Figure 1 ijms-24-08727-f001:**
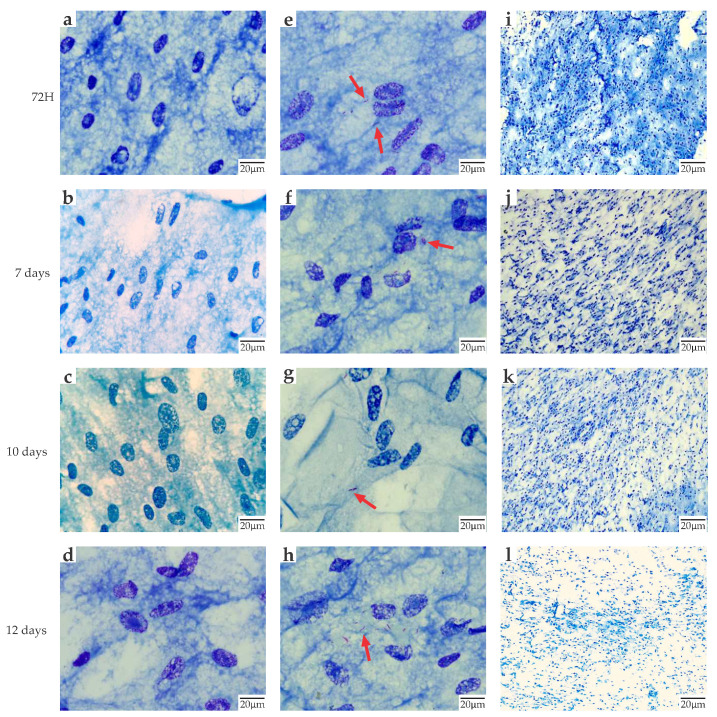
Morphology of PHSC monolayer culture infected with *M. leprae* from clinical isolates evolution: (**a**–**d**) Uninfected monolayer PHSC. light microscope 100×. Zielh Neelsen staining; (**e**–**h**) PHSC monolayer infected with *M. leprae* light microscope 100×. Zielh Neelsen staining; (**i**–**l**) Schwann cells infected with *M. leprae* light microscope 10×. Zielh Neelsen staining. Red arrow indicates the colored bacilli.

**Figure 2 ijms-24-08727-f002:**
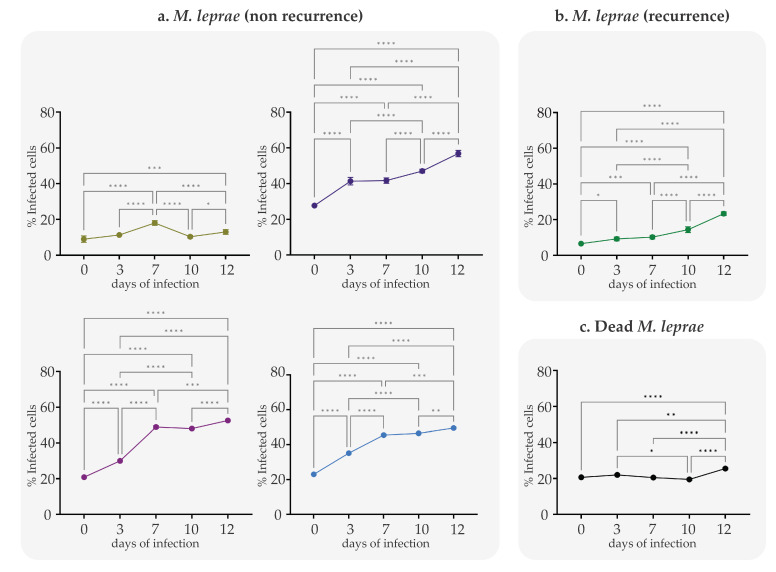
Percentage of Schwann cells infected with *M. leprae* from patients with recurrent and non-recurrent leprosy, behavior over time (**a**) SCs infected with *M. leprae* from patients with non-recurrent leprosy; (**b**) SCs infected with *M. leprae* from patients with recurrent leprosy; (**c**) SCs infected with *M. leprae* Thai strain-53 dead bacilli. The plotted values correspond to percentage of infected cells ± SEM derived from three independent infection experiments. Asterisks indicate one-way analysis of variance (ANOVA) with Tukey’s test for each time * = *p* < 0.05; ** = *p* < 0.01; *** = *p* < 0.001 and **** = *p* < 0.0001.

**Figure 3 ijms-24-08727-f003:**
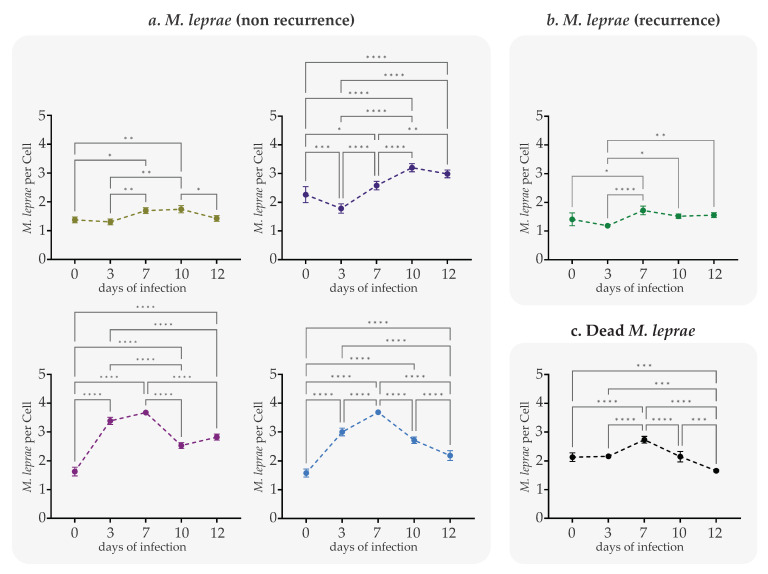
Bacilli number per Schwann cell infected with *M. leprae* from patients with recurrent and non-recurrent leprosy: (**a**) *M. leprae* from patients with non-recurrent leprosy, 100×, Zielh Neelsen staining; (**b**) *M. leprae* from patients with recurrent leprosy, 100×, Zielh Neelsen staining; (**c**) *M. leprae* Thai strain-53 dead bacilli, 100×, Zielh Neelsen staining. The plotted values correspond to number of bacilli by infected cells ± SEM derived from three independent infection experiments. Asterisks indicate one-way analysis of variance (ANOVA) with Tukey’s test for each time * = *p* < 0.05; ** = *p* < 0.01; *** = *p* < 0.001 and **** = *p* < 0.0001.

**Figure 4 ijms-24-08727-f004:**
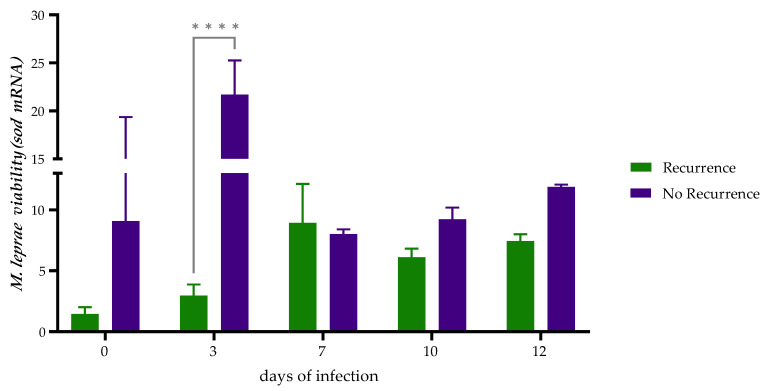
Quantification of viability of intracellular *M. leprae* clinical isolates associated and not associated with recurrent leprosy in PHSCs culture. PHSCs were infected with *M. leprae* at a MOI of 10:1. Infected cells were collected at different times of infection. Quantification was determined by the number of bacilli per cell, and the *M. leprae* viability was obtained by expression of the *sod* gene. Statistically significant differences (Mann-Whitney test; **** *= p* < 0.05).

**Figure 5 ijms-24-08727-f005:**
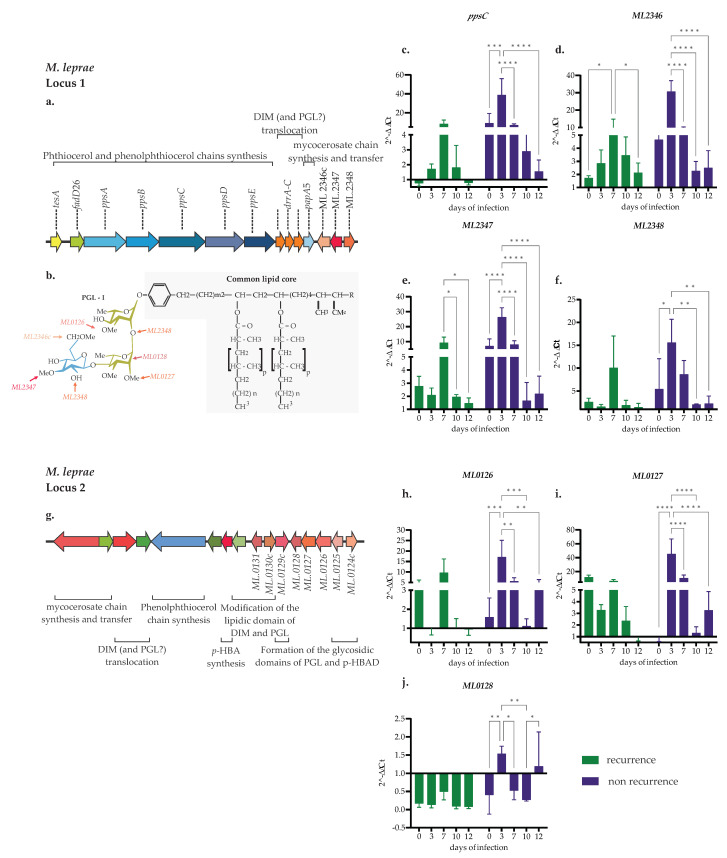
Transcriptional profile of genes involved in PGL-I synthesis of *M. leprae* clinical isolates during Schwann cell infection. (**a**) Phenolphthiocerol dimycocerosates locus 1 of *M. leprae*; (**b**) molecular structure of *M. leprae* phenolic glycolipid I; (**c**–**f**) transcription levels of locus 1 genes involved in the synthesis of PGL-I; (**g**) phenolphthiocerol dimycocerosates locus 2 of *M. leprae*; (**h**–**j**) transcription levels of locus 2 genes involved in synthesis of PGL-I. The relative transcript of the analyzed genes was normalized relative to the housekeeping gene *sod*. The values plotted correspond to the mean ± standard deviation derived from three technical replicates. Asterisks indicate two-way analysis of variance (ANOVA) with the Sidaks test * = *p* < 0.05; ** = *p* < 0.01; *** = *p* < 0.001 and **** = *p* < 0.0001.

**Table 1 ijms-24-08727-t001:** Bacteriological characterization of the *M. leprae* clinical isolates included in the study.

Code	Case	Bacillary Index	Number of Bacilli/mL	BacLight LIVE/DEAD Viability
Exp 1	Non-recurrence	1.8	1,709,403	>80%
Exp 2	Recurrence	3.0	≈1,600,000	>80%
Exp 3	Non-recurrence	5.0	27,254,012	>80%
Exp 4	Non-recurrence	2.0	106,660,872	>80%
Exp 5	Non-recurrence	2.0	62,762,831	>80%
Exp 6	Dead bacilli (Thai-53)	NA	134,451,081	0%

**Table 2 ijms-24-08727-t002:** Number of bacilli per cell quantification and infected cells per well over the study time.

Quantification	Clinical Isolate	Time 1	Time 2	Time 3	Time 4	Time 5
Number of infected cells per well	Recurrence	650	850	850	1500	2200
	Non-recurrence	2750	4116	4066	4650	5400
	dead	2066	2200	2050	1950	2550
Average number of bacilli per well	Recurrence	28,180	23,680	34,200	30,200	31,000
	Non-recurrence	45,200	35,600	51,600	64,000	59,400
	dead	42,600	43,200	54,600	42,800	33,000

**Table 3 ijms-24-08727-t003:** Sequences and characteristics of the primers designed for the detection of genes involved in the synthesis of PGL-I in *M. leprae*.

Primer	Sequence	Size	Length	Tm	% GC
*ppsC*-F	CGGAGCTAGCCGATCTCACT	127	20	64.5	60.0
*ppsC*-R	CGCACAGGATTACGCATGTT		20	60.4	50.0
ML0126-F	CTTTCGTGCGCATAATCACTG	210	21	60.6	47.6
ML0126-R	GCGACGAGATCCTCGTAATTG		21	62.6	52.4
ML0127-F	GATCTTCGCCATCTTGGACAG	178	21	62.6	52.4
ML0127-R	CTCGTATGCCTCAATGGCTTC		21	62.6	52.4
ML0128-F	CGATCCACGGTACAACAACCT	172	21	62.6	52.4
ML0128-R	TTCGATCTCGGACAGCAATTT		21	58.7	42.9
ML2346-F	ATGAAGCGTCCGAACCTGAT	104	20	60.4	50.0
ML2346-R	GGATTCGCCTCTAACGCAAC		20	62.4	55.0
ML2347-F	GATGGATCGCACTTTGGTGA	113	20	60.4	50.0
ML2347-R	CGTAGATAGCCGGGCCATAA		20	62.4	55.0
ML2348-F	GGCCTATGACGAGCTCTGCT	165	20	64.5	60.0
ML2348-R	CCGAAGCCGAAGTAGATTGG		20	62.4	55.0
*Sod-F*	CACCGTTCGGAGAGAGGTTC	192	20	64.5	60.0
*sod-R*	TCAACGAGATCCACCACACC		20	62.4	55.0

## Data Availability

The raw data supporting the conclusions of this article will be made available by the authors, without undue reservation.

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
