# Peer review of "Mycobacterium leprae’s Infective Capacity Is Associated with Activation of Genes Involved in PGL-I Biosynthesis in a Schwann Cells Infection Model"

_ijms, 2023, doi:10.3390/ijms24108727_

Round 1
Reviewer 1 Report
Chavarro-Portillo et al., in this manuscript are studying the causative agent of leprosy, Mycobacterium leprae. Even after established multi drug treatment strategy used in patients, the disease continues to infect people and continues remain a social stigma.
In this paper, the authors are more specifically studying the relationship between expression of genes responsible for synthesis of phenolic glycolipid 1 (PGL-1), a molecule that helps binding and internalization of M. leprae in Schwann cells (SCs).
In this work, the authors have used a patient biopsy samples from patients suffering from recurrent and non-recurrent M. leprae infections and in an in vitro culture setting, they have compared the kinetics of the infections and also studied the differential level of expression of genes involved in PGL-1 synthesis in non-recurring and recurring patient samples .
The biggest problem in this manuscript is the lack of enough sample size. The study is conducted on only 4 non-recurrent and 1 recurrent patient sample and there exists variability even within the sub-group. As such, the sample size is too small to draw any statistical significance and conclusion. Some controls in the experiment are also missing. For example, the negative control data for PGL-1 biosynthetic genes is missing.
This paper, as it presents has more technical replicates rather than and not biological replicates and therefore needs more samples in both the categories to make the data statistically sound.
Reviewer 2 Report
In the present study, the authors (Chavarro-Portillo et al.) studied the Mycobacterium leprae strains that survive under multidrug therapy and display a reduced metabolism, which in turn causes recurrence of leprosy. The authors assessed the infectivity in SCs of recurrent and non-recurrent M. leprae and their possible correlation with the genes involved in the (PGL-I) biosynthesis. Overall, the report suggested the lowered capacity of PGL-I production in the recurrent strain, that could affect the infection level of these strains.
The study is scientifically sound and carries high biomedical significance due to the issue of multidrug resistance in resistant strains of M. leprae.
In abstract provide future implications of the present study.
Line 127: ‘Schwann Cells’ should be written as ‘Schwann cells’
Line 40: Peripheral Nervous System (PNS) should be ‘peripheral nervous system (PNS)’
Please check all such incorrectly capitalized abbreviations.
In introduction, I recommend authors to include relevant scientific literature on gene expression (for example, Kadam, U.S., Lossie, A.C., Schulz, B., Irudayaraj, J. (2013). Gene Expression Analysis Using Conventional and Imaging Methods. In: Erdmann, V., Barciszewski, J. (eds) DNA and RNA Nanobiotechnologies in Medicine: Diagnosis and Treatment of Diseases. RNA Technologies. Springer, Berlin, Heidelberg. https://doi.org/10.1007/978-3-642-36853-0_6)
Line 147: delete full-stop from the heading
The English grammar, structure, and punctuation needs little attention.
Reviewer 3 Report
In this manuscript, Chavarro-Portillo et al. provide evidence of infection of Schwann cells with Mycobacterium leprae obtained from infected patient samples. They categorized and explained difference in infective capacity of bacteria coming from recurrent or non-recurrent infections. To be able to have confidence in this observation, this study requires more sample from recurrent infection where in the current state it is only one sample of recurrent infection compared against 4 non-recurrent samples. Apart from this, the authors should also address the following major and minor comments.
Major comments:
1. The authors describe the use of Zeihl Neelsen staining to look for intracellular bacteria in PHSC cells. An alternative method should such as membrane staining such that it can be ascertained that the bacteria are actually inside the cells.
2. Conclusions made from figure 1 are not depicted in the figure. The authors conclude that there is extensive cell loss as day 12, representative images should be added to support this observation. The same is applied to observations like cytoplasmic contraction, and increased vacuolization.
3. Figures 2 and 3: It is unclear how the authors are able to count number of infected cells and number of bacteria in each infected cell using cytoplasmic and bacterial staining. It is quite possible that in these fixed slides that bacteria are actually not infecting the cells and thus are just present on the surface of the cells. The authors should address this question in their results with explanation in the text and also supporting figures.
4. The authors also need to clarify how they are able to infect the cells with dead bacteria and then how are these dead bacteria detected with acid-fast staining.
5. Description of table 2 is incomplete, it is not clear what the values in the table represent.
Minor comments:
1. Line 52: “outer of bacillus” should be “outer layer of bacillus”
2. Line 56: “or general bacterial” should be “or general bacterial species”
3. Include scale bar in all images of Figure 1.
4. Zeihl Neelsen is incorrectly spelled as Zielh Neelsen at several places in the manuscript.
5. Figure legend of figure 2 does not match the figure.
Reviewer 4 Report
The study by Chavarro-Portillo describe the molecular and infectivity characteristics of five different clinical isolates of M. leprae that were categorized as recurrent vs non-recurrent as determined by disease presentation.
The study is interesting, however, the number of isolates is small (specially, only one isolate from recurrent disease), which hinders the study from more meaningful conclusions.
It is also not clear why the authors centered on PGL, it appears that based on ppsC expression, other phosphoglycolipids (ie.PDIM) could also be important for the differences observed. If possible, visualization of PDIM and PGL and/or total lipid extracts from cells before and after infection could help determined if there are other differences, and if the gene expression results are associated with a lipid profile difference among strains.
In addition to these considerations, a few other points are noted:
1. Table 2 should be presented in a bar graph and Standard error, or standard deviation should be noted.
2. The error bars in some of the figures are cut or missing. These should be modified to make sure that splitting the graph into two parts, does not remove the ends of the error bars.
3. Figure 5 only shows comparisons among time points for either recurrent or non-recurrent. However, in the text, the authors mention differences observed among the two types of strain. Please include comparisons in the figure or mentioned significance (p-values) in the text when comparing gene expression of recurrent vs non-recurrent strains.
Round 2
Reviewer 1 Report
I thank the authors for clearing my doubts about the use of just one recurrent sample.
One minor suggestion; Fig 2 and 3 show the individual graphs for each of the four non-recurrent clinical bacterial isolates obtained from four different patients. While table 2 suggests that the values are averaged for that group, it is not clear if in the following studies, Fig 4 and 5, if these samples were pooled or one was picked as a representative for the non-recurrent group. Information about this in the methods section might be helpful to readers.
Reviewer 3 Report
The authors should discus the limitations of the study about only having one relapse sample as opposed to 4 new infections. The authors describe that in great detail in the response to reviewers, however the readers will also benefit from this explanation. Therefore, Please add the following to the Discussion in the main text.
This work is a descriptive observational study in which it is not intended to extrapolate national or international data; Furthermore, given the characteristics of the disease, the inclusion of at least one case of leprosy relapse provides us with essential data that deserves to be analyzed and socialized. After the application of the MDT in recent years, an increase in the number of relapses has been observed; For 2016, 52 countries reported 2844 relapses out of 217971 new cases; in 2017, 3192 cases out of 211182; in 2018 and 2019, 3361 and 3893 cases out of 208641 and 202185 new cases, respectively (WHO, 2019; WHO, 2020). When reviewing the worldwide incidence of relapse cases in leprosy, it is clear that it is low (1.16 per 1000 py (95% CI = 0.5915–2.076)) (Nery JAC, 2021 and Guerrero et al., 2012) and that it is necessary to follow one thousand leprosy cases to obtain less than two cases of relapse of the disease and that the period of follow-up could be long and variable, because of censoring and loss to follow-up; This makes it challenging to include more cases of leprosy relapses in the present study (Nery JAC, 2021 and WHO 2016) also considering that Colombia has 400 new
cases of leprosy per year, (WHO, 2019).
World Health Organization. Global leprosy update, 2018: need for early case detection. Wkly Epidemiol Rec
2019;94, 389–412. Available from: https://www.who.int/wer/2019/wer9435_36/en/.
World Health Organization. Global leprosy (Hansen disease) update, 2019. Wkly Epidemiol Rec 2020;95, 417–
440. Available from: https://www.who.int/wer/2020/wer9536/en/.
Nery JAC, Sales AM, Hacker MAVB, Moraes MO, Maia RC, et al. (2021) Low rate of relapse after twelve-dose multidrug therapy for hansen’s disease: A 20-year cohort study in a brazilian reference center. PLOS Neglected Tropical Diseases 15(5): e0009382. https://doi.org/10.1371/journal.pntd.0009382
Guerrero-Guerrero, M.I.; Muvdi-Arenas, S.; Leon-Franco, C.I. Relapses in Multibacillary Leprosy Patients: A Retrospective Cohort of 11 Years in Colombia. Leprosy review 2012, 83, 247–260.
World Health Organization. Global leprosy strategy 2016–2020: Accelerating towards a leprosy-free world— operational manual. WHO Regional Office for South-East Asia. Delhi, 2016. Available from: http://www.who.int/iris/handle/10665/250119.
